# Silkworm Pupae Function as Efficient Producers of Recombinant Glycoproteins with Stable-Isotope Labeling

**DOI:** 10.3390/biom10111482

**Published:** 2020-10-26

**Authors:** Hirokazu Yagi, Saeko Yanaka, Rina Yogo, Akari Ikeda, Masayoshi Onitsuka, Toshio Yamazaki, Tatsuya Kato, Enoch Y. Park, Jun Yokoyama, Koichi Kato

**Affiliations:** 1Graduate School of Pharmaceutical Sciences, Nagoya City University, 3-1 Tanabe-dori, Mizuho-ku, Nagoya 467-8603, Japan; hyagi@phar.nagoya-cu.ac.jp (H.Y.); saeko-yanaka@ims.ac.jp (S.Y.); yogo@ims.ac.jp (R.Y.); 2Exploratory Research Center on Life and Living Systems (ExCELLS) and Institute for Molecular Science (IMS), National Institutes of Natural Sciences, 5-1 Higashiyama, Myodaiji-cho, Okazaki 444-8787, Japan; 3Taiyo Nippon Sanso Corporation, SI Innovation Center, 2008-2 Wada, Tama, Tokyo 206-0001, Japan; ikedaa.qnm@tn-sanso.co.jp (A.I.); yokoyamaj.qrb@tn-sanso.co.jp (J.Y.); 4Graduate School of Technology, Industrial and Social Sciences, Tokushima University, Minamijosanjima-cho 2-1, Tokushima 770-8513, Japan; onitsuka@tokushima-u.ac.jp; 5SPring-8 Center RIKEN, 1-7-22 Suehiro-cho, Tsurumi-ku, Yokohama City, Kanagawa 230-0045, Japan; toshio.yamazaki@riken.jp; 6Laboratory of Biotechnology, Research Institute of Green Science and Technology, Shizuoka University, 836 Ohya Suruga-ku, Shizuoka 422-8529, Japan; kato.tatsuya@shizuoka.ac.jp (T.K.); park.enoch@shizuoka.ac.jp (E.Y.P.)

**Keywords:** silkworm pupa, isotope labeling, recombinant glycoprotein, artificial diet, immunoglobulin G

## Abstract

Baculovirus-infected silkworms are promising bioreactors for producing recombinant glycoproteins, including antibodies. Previously, we developed a method for isotope labeling of glycoproteins for nuclear magnetic resonance (NMR) studies using silkworm larvae reared on an artificial diet containing ^15^N-labeled yeast crude protein extract. Here, we further develop this method by introducing a technique for the expression of isotope-labeled glycoproteins by silkworm pupae, which has several potential advantages relative to larvae-based techniques in terms of production yield, ease of handling, and storage. Here, we fed fifth instar larvae an artificial diet with an optimized composition containing [methyl-^13^C]methionine, leading to pupation. Nine-day-old pupae were then injected with recombinant *Bombyx mori* nucleopolyhedrovirus (BmNPV) bacmid for expression of recombinant human immunoglobulin G (IgG). From the whole-body homogenates of pupae, 0.35 mg/pupa of IgG was harvested, which is a yield that is five times higher than can be obtained from larvae. Recombinant IgG, thus prepared, exhibited mainly three kinds of pauci-mannose-type oligosaccharides and had a ^13^C-enrichment ratio of approximately 80%. This enabled selective observation of NMR signals originating from the methionyl methyl group of IgG, confirming its conformational integrity. These data demonstrate the utility of silkworm pupae as factories for producing recombinant glycoproteins with amino-acid-selective isotope labeling.

## 1. Introduction

Recombinant proteins are widely used as research tools in every aspect of life science and, moreover, as pharmaceuticals, catalysts, and biomaterials in industrial applications. To date, a variety of options are available for producing recombinant proteins, especially in terms of choice of expression vehicles, which include bacteria, yeast, plants, insects, mammals, and even cell-free systems, according to purposes [1,2]. In making one’s choice, production cost, yield, and ease of handling both in culture and during genetic manipulation are generally considered.

Recombinant proteins are often expressed with modifications, as exemplified by mutational modifications such as tagging, truncation, and/or amino-acid substitution and by post-translational modifications such as glycosylation. Isotope enrichment is also a useful modification, and is especially important for recombinant proteins subjected to structural analyses such as nuclear magnetic resonance (NMR) spectroscopy. At the laboratory level, *Escherichia coli* is one of the most widely used expression vehicles because of its innate advantages, i.e., fast growth, well-characterized genetics, cost-effectiveness, and high yield [3,4]. However, this conventional expression vehicle is not equipped with a glycosylation system. Therefore, eukaryotic vehicles are alternatively used for production of recombinant glycoproteins [1,2].

Several groups have developed stable-isotope labeling protocols to produce recombinant glycoproteins using eukaryotic expression systems [5,6,7,8,9,10,11]. Our group has reported protocols for the stable-isotope labeling of recombinant glycoproteins by transgenic tobacco, and silkworm larvae as well as mammalian cells, using immunoglobulin G (IgG) glycoproteins as models [12,13,14,15]. Recombinant IgG glycoproteins have been used as therapeutic antibodies and NMR spectroscopy has been considered as a useful tool for assessing their higher-order structural integrity [16].

In this context, a baculovirus-infected silkworm system serves as a highly efficient and cost-effective bioreactor for the production of heterologous glycoproteins including IgGs [17,18,19]. In this system, a *Bombyx mori* nucleopolyhedrovirus (BmNPV) bacmid containing the gene sequences of target proteins is injected into a silkworm larva. Several attempts have been made to improve production yield and to remodel functional glycosylation of *B. mori*-derived IgG glycoproteins. In our previous paper, we demonstrated that recombinant IgG with approximately 80% ^15^N enrichment could be produced in maintaining structural integrity by silkworm larvae reared with an isotope-labeled artificial diet [14].

Here, we explore the possibility of exploiting silkworm pupae—instead of larvae—to produce isotope-labeled recombinant glycoproteins. Pupae possess potential advantages relative to larvae [17]. Pupae do not require feeding and do not move, and so are easier to maintain than larvae. Unlike larvae, pupae can be stored at a temperature of 4 °C for up to two weeks prior to bacmid injection, and can also produce higher yields of recombinant proteins than larvae.

In this study, we attempt to optimize a silkworm rearing condition so as to maximize production yield of recombinant IgG by the use of *B.mori* pupae. Since we have already achieved full ^15^N labeling [14], our trial in this study is designed to perform amino-acid selective ^13^C labeling of IgG.

## 2. Materials and Methods

### 2.1. Artificial Diets for Silkworm Rearing

Artificial diets used for rearing silkworms consisted of the components described previously with modified compositions [20,21,22] (Table 1). Here, an artificial diet containing 40% amino acid and 10% mulberry leaf powder is termed as A40M10. Similar notation will be used for other compositions of artificial diets. For amino-acid-selective isotope labeling, a protein mixture derived from the yeast strain, *Candida utilis* NBRC 0396, was replaced by an amino-acid mixture (Table 2) containing [methyl-^13^C]methionine.

### 2.2. Expression and Purification of Recombinant IgG

Recombinant human IgG1 directed against bovine serum albumin [23] was produced in silkworms using rBmNPV bacmid systems using protocols described previously [14,18,19,24]. Fifth-instar hybrid Kinsyu×Syowa silkworm larvae (Funakoshi Co., Ltd. Tokyo, Japan) were fed on an artificial diet every 12 h, and were kept at 27 °C with a 12 h/12 h light/dark cycle. To produce recombinant IgG1, genes for the heavy and light chains were cloned into a cysteine protease-deficient and chitinase-deficient *B. mori* nucleopolyhedrovirus (BmNPV-*CP^-^*-*Chi^-^*) bacmid, yielding a recombinant virus (BmNPV bacmid/29IJ6 IgG) used to infect silkworms. With proper feeding, the fifth-instar larvae pupated typically nine days later, which were subsequently incubated at 27 °C in dark for another nine days. To express recombinant IgG1, BmNPV bacmid/29IJ6 IgG was injected into nine-day-old pupae or two-day-old fifth instar larvae, together with 1,2-dimyristyloxypropyl-3-dimethyl-hydroxyethyl ammonium bromide (DMRIE-C, Life Technologies Japan, Tokyo, Japan). Bacmid-injected larvae and pupae were maintained at 27 °C for 6 and 7 days, respectively, for protein expression. Timeline and procedure of the protein expression in silkworm pupae are summarized in Figure 1.

Purification of IgG was performed as previously described [14,18]. The bacmid-injected pupae were homogenized with Tris-buffered saline containing Tween-20, and were then centrifuged (20,000× *g*, 40 min) after which the sediments were removed. From the bacmid-injected larvae, the hemolymph was collected and centrifuged (19,000× *g* for 30 min). The supernatant was filtered using a 0.45-μm nitrocellulose membrane filter (Merck Millipore, Burlington, MA, USA) and then loaded onto an Affi-Gel protein A column (GE Healthcare Life-Sciences, Marlborough, MA, USA). IgG was further purified by gel-filtration chromatography using a Superose 12 column (GE Healthcare Life-Sciences, Marlborough, MA, USA).

Chinese hamster ovary (CHO) cell line producing rituximab, anti-CD20 mouse-human chimeric IgG1, was prepared in-house. Briefly, heavy-chain and light-chain genes of rituximab were ligated into the Mammalian PowerExpress system^®^ vector (Toyobo, Otsu, Japan). Suspension-adapted CHO-K1 cells (ATCC CRL-9618) were transfected with the vector and the stable cell lines were established by single colony picking. Isotope labeling of rituximab was made as described previously with slight modifications [12,13]. For methionine-specific ^13^C labeling, the CHO cell line was cultivated with a modified Nissui NYSF 404 medium (supplemented with 2% dialyzed fetal bovine serum and 7.5 μg/mL puromycin) in which L-methionine was replaced by [methyl-^13^C]methionine. Following the growth of these cells, IgG proteins were purified from the supernatant of the medium using protein A affinity chromatography and gel filtration as described above. In this study, 2.5 mg of recombinant IgG glycoprotein was subjected to NMR measurements, which was produced with 7 pupae fed with 82 g of artificial diet containing 200 mg of [methyl-^13^C]methionine.

### 2.3. Limited Proteolysis of IgG

The Fab and Fc fragments of IgG were prepared as previously described [25,26]. IgG was cleaved by papain digestion, performed at 37 °C for 12 h in 75 mM sodium phosphate buffer [pH 7.0, containing 75 mM NaCl and 2 mM ethylenediaminetetraacetic acid (EDTA)]. The IgG concentration was 10 mg/mL and the papain: IgG ratio was 1:50 (*w*/*w*). The digestion products were loaded onto an Affi-Gel protein A column. Fc and Fab fragments were then collected from the bound and unbound fractions, respectively. Both fragments were further purified by a Superose 12 gel-filtration column. Purity of each of the protein preparations was checked by sodium dodecyl sulfate-polyacrylamide gel electrophoresis (SDS-PAGE).

### 2.4. Mass Spectrometric Measurement of Stable-Isotope Enrichment

Prior to amino-acid analysis, samples of isotope-labeled artificial diet, crude protein extract, and IgG purified from silkworm pupae were hydrolyzed by HCl for 24 h at 110 °C. After cooling to room temperature, the hydrolyzate was filtered through a polytetraflouroethylene (PTFE) membrane filter (0.45-μm pore size, Millipore, Burlington, MA, USA). The solution was transferred into an eggplant-shaped flask, and unreacted HCl was removed by evaporation under a vacuum at 40 °C. The dried residue was dissolved in Milli-Q water. Liquid chromatography-mass spectrometry (LC-MS) system used in this study was composed of an ACQUITY UPLC (Waters Corporation, Milford, MA, USA) coupled to a Xevo G2-XS QTOF mass spectrometer (Waters Corporation) with a heated electrospray ionization (ESI) source. Amino acids were separated on a Discovery HSF5 column (Merck, Kenilworth, NJ, USA) with dimensions of 2.1 i.d.×150 mm and a particle size of 3-μm. A set of mobile phases was used: 0.1% (*v*/*v*) formic acid (A) and 0.1% (*v*/*v*) formic acid in acetonitrile (B). The gradient conditions were as follows: *t* = 0–5 min, 0% B, *t* = 5–15 min, 0–40% B, *t* = 15–16 min, 40–100% B, *t* = 16–20 min, 100% B, *t* = 20–20.1 min, 100–0% B, *t* = 20.1–25 min, 0% B. The flow rate was set at 0.25 mL/min and the column oven temperature was 45 °C. The injection volume was 1 μL. The ESI-MS conditions were as follows: Source Temperature, 120 °C, Desolvation Temperature, 450 °C, Desolvation Gas Flow, 800 L/h, Cone Gas Flow, 50 L/h, and capillary voltage, 1.5 kV for positive mode.

### 2.5. Glycosylation Profiling

*N*-glycosylation profiling of IgG was performed based on the high-performance liquid chromatography (HPLC) mapping technique described previously [27]. Isotope-labeled IgG-Fc glycoprotein (0.2 mg) was used as a starting material. *N*-glycans of the pupa-derived IgG glycoprotein were released by hydrazinolysis [28,29], tagged with 2-aminopyridine, and loaded onto an amide column (Tosoh Co., Tokyo, Japan). Next, each fraction separated on the amide column was subjected to an octadecylsilyl ODS column (Shimadzu Co., Kyoto, Japan). The elution time on each HPLC column was expressed in the glucose units (G.U.), in reference to the pyridylamino (PA) derivatized isomalto-oligosaccharide mixture. Structural identification was based on the elution position on the HPLC column in the GALAXY database (http://www.glycoanalysis.info) [27].

### 2.6. NMR Measurements

For NMR measurements, Fab/Fc fragments and full-length IgG glycoproteins were dissolved in 0.5 mL of 5 mM sodium phosphate buffer [pH 6.0, containing 50 mM NaCl and 10% (*v*/*v*) D_2_O] at a protein concentration of 10 mg/mL. Two-dimensional methyl-transverse relaxation optimized spectroscopy(TROSY) spectral data were acquired at 37 °C using an AVANCE 800 spectrometer equipped with a cryogenic probe (Bruker BioSpin, Fällanden, Switzerland). Assignments for the methionyl methyl resonances of Fc were made based on the previously reported backbone assignments [30] by analyzing nuclear Overhauser effect (NOE) connectivities observed using AVANCE 800 and AVANCEIII 900 spectrometers equipped with cryogenic probes (Bruker BioSpin, Fällanden, Switzerland). Chemical shifts of ^1^H were referenced to 4,4-dimethyl-4-silapentane-1-sulfonic acid (0 ppm), and ^13^C chemical shifts were referenced indirectly using the gyromagnetic ratios of ^13^C and ^1^H (*γ*^13^C/*γ*^1^H = 0.25144952). All NMR data were processed using NMR Pipe [31] and were analyzed using CCPNMR [32].

## 3. Results and Discussion

### 3.1. Optimization of Artificial Diet Composition

In our previous study, ^15^N-labeled IgG was produced by silkworm larvae fed the artificial diet A40M10 in which amino acids were supplied as yeast crude protein extract [14]. However, the larvae fed this artificial diet did not grow enough to pupate. In addition, amino-acid selective labeling is difficult in this protocol. To solve these problems, we adopted the A30M10 and A20M20 artificial diets in which the yeast extract was replaced by an amino-acid mixture. These recipes allowed the larvae to grow into pupae at day 9 (Figure 2). Comparison of the two growth curves indicate that a higher amount of mulberry powder enhances larval appetite and growth. Hence, we used A20M20 for expression of recombinant IgG.

### 3.2. Production of IgG in Silkworm Pupae

IgG production yields were compared between larvae and pupae from identical A20M20-fed rearing environments. The only difference between treatments was the time point of bacmid injection, i.e., 2-day-old fifth instar larvae and 9-day-old pupae. Larval growth significantly slowed after bacmid injection (Appendix A). Recombinant IgG yields purified from the hemolymph fluids of larvae and whole-body homogenates of pupae were 0.07 mg/larva and 0.35 mg/pupa, respectively. Thus, pupae extracts showed a five-fold greater yield than larval extracts.

BmNPV infects blood cells and fat body cells in silkworms. Pupal body tissues primarily consist of fat body cells, which can express recombinant proteins. For this reason, recombinant IgG can be harvested from pupal whole-body homogenates. In contrast, because the larval midgut contains a large amount of proteases, it is necessary to extract body fluid prior to recombinant protein purification [33,34,35]. It is possible that this is the reason for the higher recombinant IgG yield of pupae relative to larvae. There have been several reports of glycoprotein production in silkworm pupae including porcine lactoferrin (0.2 mg/pupa) [36] and human granulocyte macrophage colony stimulating factor (0.1 mg/pupa) [37], suggesting that our method can be applicable for isotope-labeling of other proteins in high yields.

Recombinant IgG purified from pupae was subjected to HPLC-based glycosylation profiling (Appendix A and Table 3). The *N*-glycans mainly consisted of three kinds of pauci-mannose-type oligosaccharides [Manα1-6Manβ1-4GlcNAcβ1-4(Fucα1-6)GlcNAc, Manα1-6(Manα1-3)Manβ1-4GlcNAcβ1-4(Fucα1-6)GlcNAc, and Manα1-6(GlcNAcβ1-2Manα1-3)Manβ1-4GlcNAcβ1-4(Fucα1-6)GlcNAc]. These results are consistent with those of previous reports [18,24], confirming that the pupa-derived IgG retained the nonreducing terminal GlcNAcβ1-2 moiety, unlike the larva-derived IgG. This difference may be due to reduced β-*N*-acetylhexosaminidase activity in pupae.

### 3.3. Characterization of ^13^C-Labeled IgG

Using the A20M20 diet containing [methyl-^13^C]methionine, we performed metabolic ^13^C labeling of the recombinant IgG glycoprotein produced by pupae (Figure 3). LC-MS analyses revealed that ^13^C enrichment degrees of methionine were 99% in the A20M20 diet used for rearing and approximately 80% in the crude protein fractions and recombinant IgG harvested from the pupae (Figure 4). Since we started rearing silkworm larvae with the isotope-labeled artificial diet at their fifth instar stage, the isotope dilution is likely to be ascribed to pre-existing unlabeled amino acids in the silkworm body. An earlier start of rearing with the isotope-labeled diet would improve ^13^C labeling efficiency but presumably result in a lower production yield. Taking account of such a trade-off, we suppose that an 80% enrichment is a good compromise for measuring heteronuclear NMR spectra. Our protocol enabled selective observation of NMR signals originating from IgG methionyl methyl groups (Figure 5).

The recombinant IgG contains three and two methionine residues in the Fab and Fc regions, respectively [23]. This was confirmed by spectral data for the Fab and Fc fragments, which exhibited additivity so as to reproduce the full-length IgG spectrum (Figure 5B,C). Furthermore, the Fc spectrum was compared with that of an authentic human IgG1-Fc preparation produced in CHO cells, thereby confirming the conformational integrity of the Fc region (Figure 5C,D).

These data indicate that silkworm pupae are capable of producing stable-isotope labeled recombinant IgG in much higher yields than larvae. We have previously demonstrated that antibody production yield in silkworm can be improved by a factor 5 upon co-expression with molecular chaperones [18]. When compared with the mammalian expression systems taking this into account, the cost for producing the isotope-labeled IgG glycoproteins are comparable in silkworm pupae, which, however, do not require any expensive equipment such as the CO_2_ incubator. Therefore, the total cost will be less expensive in the silkworm-based method, which also does not take up much space, enabling production of different proteins with different labeling modes in parallel. In terms of the time period for obtaining the products, both the mammalian and pupa systems require approximately three to four weeks from the start of culture with isotope-labelled metabolic precursors to harvest. However, it generally takes a couple of months to establish a mammalian cell line that can produce a recombinant antibody at a high expression level. Thus, the pupa expression system is excellent in terms of flexibility and efficiency in contrast to large-scale culture of mammalian cells.

## 4. Conclusions

In this study, we demonstrate the utility of silkworm pupae as factories for producing isotope-labeled recombinant glycoproteins. This is a significant improvement over previous work using silkworm larvae since pupae are easier to handle and store and show an increased IgG yield. Moreover, the protocol developed here enables amino-acid selective labeling of glycoproteins, which complements the non-selective labeling technique we established previously [14]. Efficiency of amino-acid selective labeling depends on metabolic pathways. In silkworm, essential amino acids are arginine, histidine, isoleucine, leucine, lysine, methionine, phenylalanine, threonine, tryptophan, and valine, while aspartic and glutamic acids are interconverted by transaminases in larval tissues [38]. These have to be taken into account in order to improve labeling efficiency.

Recombinant IgG glycoproteins are currently used as biopharmaceuticals [39,40]. NMR is expected to offer a useful tool for assessing conformational integrity of therapeutic antibodies. In particular, NMR signals originating from methyl groups can be useful spectroscopic probes for monitoring local conformations in larger glycoproteins, such as IgG [41,42]. Our pupa-based protocol can be useful for preparing a series of glycoprotein samples in which different amino-acid residues are selectively labeled because it can save time, space, and effort relative to those using larvae or other eukaryotic expression vehicles. Several attempts have been made not only for increasing yields of recombinant glycoproteins produced by silkworms, e.g., by co-expression with molecular chaperones [18] but also to engineer their glycoforms, e.g., by co-expression with human glycosyltransferases [24]. Our pupae-based approach combined with these techniques opens the possibility of recombinant glycoproteins with functional modifications.

## Figures and Tables

**Figure 1 biomolecules-10-01482-f001:**
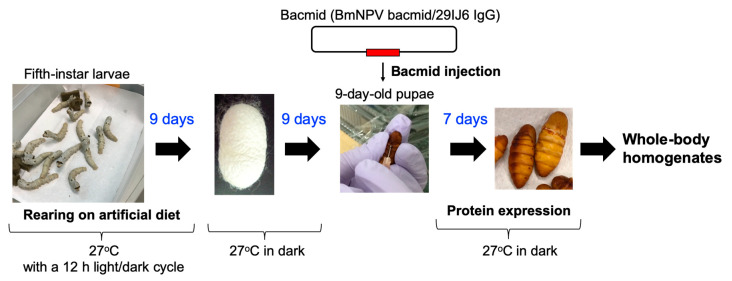
Timeline and procedure of the protein expression in silkworm pupae.

**Figure 2 biomolecules-10-01482-f002:**
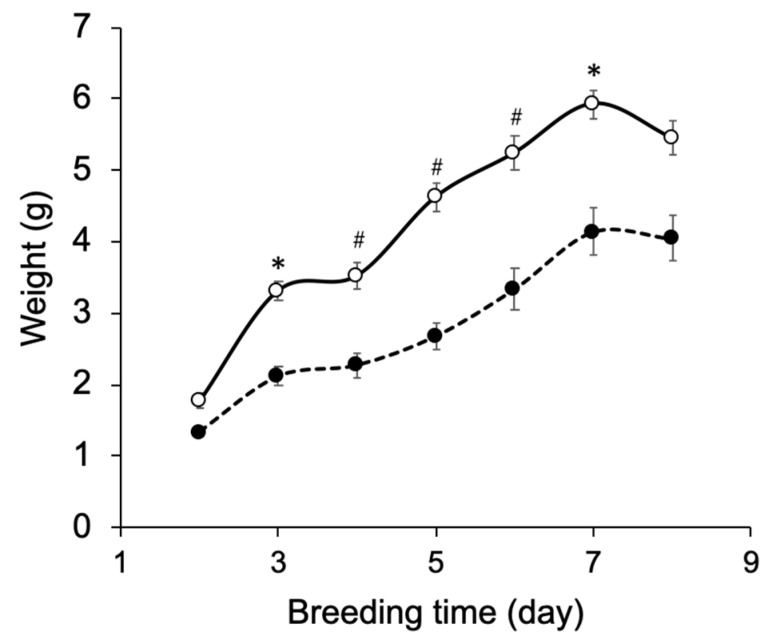
Growth curve of silkworm larvae reared on artificial diets. Silkworm larvae (*n* = 10) were reared on the artificial diets A20M20 (open circle) and A30M10 (closed circle) from the fifth-instar larval stage. Mean ± s.e.m. values are shown, * *p* < 0.001 and *^#^ p* < 0.0001 by Student’s *t*-test.

**Figure 3 biomolecules-10-01482-f003:**
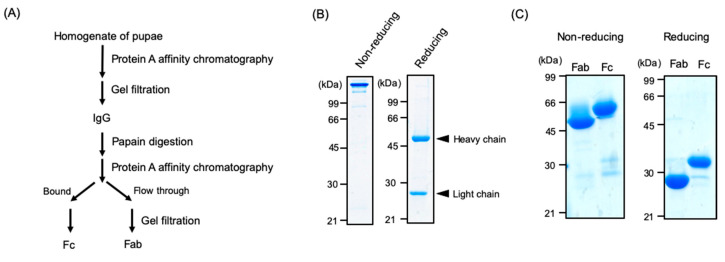
Preparation of IgG and its proteolytic fragments from silkworm pupae. (**A**) Scheme for purification and limited proteolysis of IgG. SDS-PAGE profiles of (**B**) IgG and (**C**) its Fab and Fc fragments from the pupae under non-reducing and reducing conditions. Arrowheads indicate the positions of heavy or light chains.

**Figure 4 biomolecules-10-01482-f004:**
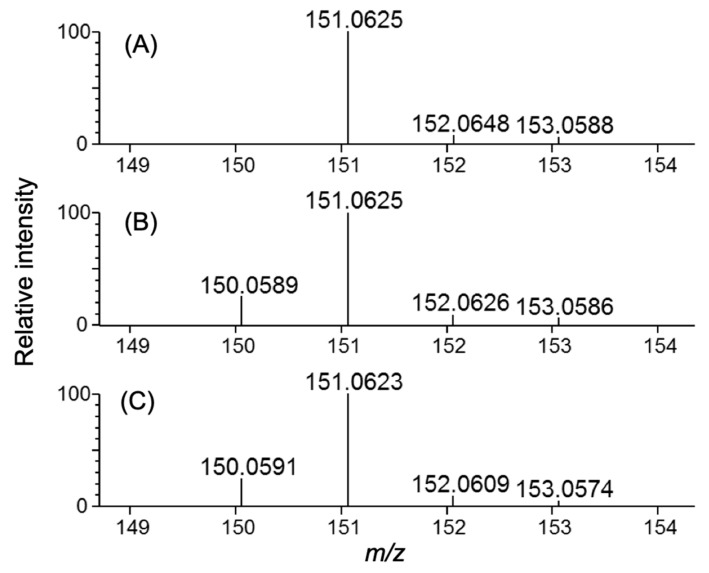
Methionine LC-MS spectra of isotopically labeled samples. Samples shown are: (**A**) artificial diet A20M20 containing [methyl-^13^C]methionine used for metabolic labeling, (**B**) homogenate, and (**C**) purified IgG originating from silkworm pupae reared on the ^13^C-labeled artificial diet.

**Figure 5 biomolecules-10-01482-f005:**
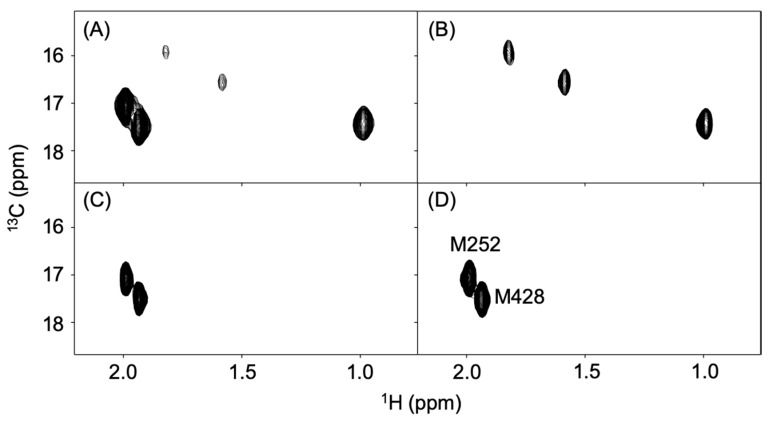
Methyl-TROSY spectra of recombinant IgG and its proteolytic fragments labeled with ^13^C at the methionyl methyl groups. (**A**) full-length IgG produced in silkworm pupae, (**B**) Fab, and (**C**) Fc fragments of pupae-expressed IgG and (**D**) Fc fragment of CHO-expressed IgG.

**Table 1 biomolecules-10-01482-t001:** Compositions of artificial diets used to cultivate silkworm larvae.

Substances	% of Dry Diet		
Artificial diet	A40M10	A30M10	A20M20
Yeast protein	40.0	- ^4^	- ^4^
Amino acid mixture	- ^4^	30.0	20.0
Mulberrry leaf powder	10.0	10.0	20.0
Glucose	12.0	15.0	15.0
Soybean oil	3.0	3.0	3.0
Phytosterol	0.3	0.3	0.3
Ascorbic acid	2.0	2.0	2.0
Sorbic acid	0.2	0.2	0.2
Agar	10.0	10.0	10.0
Salt mixture ^1^	4.2	4.2	4.2
Vitamin B mixture ^2^	0.4	0.4	0.4
Cellulose powder	17.9	24.9	24.9
(Total)	100.0	100.0	100.0
Antiseptics ^3^	Added	Added	Added
Distilled water	3 mL/g diet	3 mL/g diet	3 mL/g diet

^1^ See Reference [22]. ^2^ See Reference [21]. ^3^ Antiseptics consisted of chloramphenicol (0.015% in diet) and propionic acid (0.75% in diet). ^4^ Absent compound.

**Table 2 biomolecules-10-01482-t002:** Amino-acid compositions of the yeast protein/amino-acid mixture contained in the artificial diets.

	Yeast Protein	Amino Acid Mixture ^1^
Amino Acids	mol%	
Alanine	9.30	10.75
Arginine	4.23	3.90
Aspartic acid	11.09	9.48
Cysteine		2.82
Glutamic acid	11.45	9.71
Glycine	7.23	8.20
Histidine	2.10	1.78
Isoleucine	5.20	5.74
Leucine	8.55	8.86
Lysine	7.45	4.49
Methionine	1.39	2.28
Phenylalanine	4.06	4.56
Proline	4.21	5.94
Serine	7.11	4.56
Threonine	6.60	5.74
Tryptophan		2.68
Tyrosine	3.29	1.51
Valine	6.77	7.01
Total	100.0	100.0

^1^ See Reference [20].

**Table 3 biomolecules-10-01482-t003:** *N*-glycosylation profile of IgG expressed by silkworm pupae reared on artificial diet A20M20.

G.U. (ODS)	G.U. (Amide)	Relative Quantity (%)	Structure
10.4	3.5	36.1	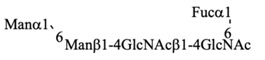
10.3	4.8	41.1	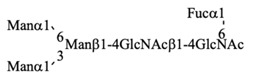
10.3	5.2	16.6	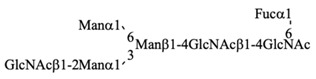
		6.2	others

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
