# Peer review of "Silkworm Pupae Function as Efficient Producers of Recombinant Glycoproteins with Stable-Isotope Labeling"

_biomolecules, 2020, doi:10.3390/biom10111482_

Round 1

Reviewer 1 Report

NMR is one of the most powerful and versatile techniques to study the structural and functional properties of the biological macromolecules, in particular of proteins and protein/ligand complexes. One of the major limitation of the technic is the necessity to enrich the studied object in stable isotopes (i.e. replacing the 14N, 12C by 15N, 13C and sometime the 1H by 2H). For this reason, NMR is mainly restricted to protein produced in Escherichia coli where labelling can be in general easily achieved at low cost. But many important proteins have to be produced in mammalian and insect cells to be functional, or could be produced in E. coli but would deserve to be directly studied (and accordingly produced) in their eukaryotic cellular environment.
Many different options have been explored to solve the question of labelled protein production in eukaryotic systems. Several robust and cost effective systems are now available, but they still require more expertise and the price is still much higher than for E. coli productions. Accordingly there is room for other solutions.
Here, the authors propose to use Silkworm pupae to produce labeled proteins, in the continuity of a previous work where they used the larvae of the same moth for the same purpose. They focus on amino-acid selective labelling of antibodies, with the rational that the production of therapeutic antibodies is a growing issue and that NMR is a well adapted quality control tool.
The experiments the authors show are convincingly performed and their results appear conform to what could be expected. Nevertheless, the article is globally frustrating. The main question that a reader would like to see answered is whether it will be interesting for her/him to use this method instead of another one when confronted to the problem of producing a labelled protein that require an eukaryotic expression system. Several points should be explored: the ease of implementation and use of the method, its robustness and generality, and finally, its cost.
The authors claim that the method is easy to set up, but considering that most reader are not familiar with larvae/pupae culture, more details concerning the equipment and the handling requirement would be welcome.
In addition, the authors explain that the pupae do not allow for uniform labelling and can only be used for specific labelling, which is a strong limitation of the method, considering in particular that other very efficient methods has been proposed for this purpose. In addition, the authors only present the case of 13CH3-methionine labelling, which is the easiest and the cheapest to perform (there exist several mediums depleted in methionine for the culture of insect and mammalian cells that have been optimised for the production of eukaryotic proteins, in particular for the production of antibodies, and that could be readily used for the production of 13CH3-Met labeled samples). Even in the context of specific amino-acid labelling, several questions should be addressed by the authors to give their method some generality: i) the level of transaminase activity, that could impact the possibility to perform 15N-specific labelling and ii) the level of scrambling between amino-acid side-chains and the interplay between the neo-synthesis of amino-acid from the glucose and their direct incorporation from the medium that could affect both the specificity of the labelling and its efficiency. For this, an example with one or two other 15N,13C-labeled amino-acids would be important.
Finally, the most important point is likely the cost of the method. The authors should discuss the cost of the production of a typical NMR sample (100 uM in 200 ul) in different situations (a well expressed protein as an antibody, a "normally" expressed protein). It would be interesting to have one or two examples as most reader will have no idea of what could be expected from the system. This cost should be compared to what could be expected by using other systems, as insect cells, mammalian cells or cell-free systems.
In conclusion, I think it would be interesting to find in such a paper:
- The yield results obtained with one or two additional protein, to have an idea of the expression level that could be obtained with more than one protein. The expression yield should be reported to the amount of medium needed to perform the culture.
- The spectra obtained with several 15N-13C-double labelled amino-acids and the comparison of what should be expected in the absence of any transamination process or side-chain scrambling. It also would be important to have a measure of the 15N and 13C labelling level for these amino-acids. It would also be important to have a discussion of the labelling process in the context of what is known about the silkworm amino-acid metabolism.
- A more complete discussion of the advantages/disadvantages and cost of the system with respect with those that have been already published.

Author Response

Response to reviewer 1

The authors claim that the method is easy to set up, but considering that most reader are not familiar with larvae/pupae culture, more details concerning the equipment and the handling requirement would be welcome. 

We concur with the reviewer. Accordingly, we added a new figure showing the procedures of protein expression in silkworm pupae with pictures (p.4).

In addition, the authors explain that the pupae do not allow for uniform labelling and can only be used for specific labelling, which is a strong limitation of the method, considering in particular that other very efficient methods has been proposed for this purpose. In addition, the authors only present the case of 13CH3-methionine labelling, which is the easiest and the cheapest to perform (there exist several mediums depleted in methionine for the culture of insect and mammalian cells that have been optimised for the production of eukaryotic proteins, in particular for the production of antibodies, and that could be readily used for the production of 13CH3-Met labeled samples). Even in the context of specific amino-acid labelling, several questions should be addressed by the authors to give their method some generality: i) the level of transaminase activity, that could impact the possibility to perform 15N-specific labelling and ii) the level of scrambling between amino-acid side-chains and the interplay between the neo-synthesis of amino-acid from the glucose and their direct incorporation from the medium that could affect both the specificity of the labelling and its efficiency. For this, an example with one or two other 15N,13C-labeled amino-acids would be important.

Regarding the isotope labeling mode, we are afraid that our intentions may not be properly understood by the reviewer. We have never mentioned that the pupae do not allow for uniform labelling and can only be used for specific labelling. Indeed, we demonstrated that it is possible to perform uniform labeling with silkworm larvae in our previous paper [Yagi et al. (2015) J. Biomol. NMR, 62, 157-167] and there is no rational reason why this cannot be applied to the pupa-based method. In view this situation, we herein test the feasibility of amino-acid selective labeling in silkworm, rather than uniform labeling, using [methyl-13C]methionine simply as a test model. Needless to say, the efficiency of selective labeling depends on metabolic pathways and transamination often precludes specific 15N labeling. However, the main subject of this study is to examine the potential advantages of pupae in terms of production yield and ease of handling as compared with the larva-based system, which we previously established. Although the isotope labeling efficiency could be improved by engineering of the silkworm metabolic pathway, we are afraid that this is beyond the scope of this study. We modified the descriptions in the text so as to avoid possible misunderstanding of our intentions (p.2, lines 74-75).

Finally, the most important point is likely the cost of the method. The authors should discuss the cost of the production of a typical NMR sample (100 uM in 200 ul) in different situations (a well expressed protein as an antibody, a "normally" expressed protein). It would be interesting to have one or two examples as most reader will have no idea of what could be expected from the system. This cost should be compared to what could be expected by using other systems, as insect cells, mammalian cells or cell-free systems.

We thank the reviewer for the constructive comments. The cost issue will be addressed in our reply to the last comments raised by this reviewer (see below).

In conclusion, I think it would be interesting to find in such a paper:

- The yield results obtained with one or two additional protein, to have an idea of the expression level that could be obtained with more than one protein. The expression yield should be reported to the amount of medium needed to perform the culture.

We are afraid that we have to say that it is an excessive demand and difficult to establish another expression system within a reasonable timeframe. However, there have been several reports of glycoprotein production in silkworm pupae including porcine lactoferrin (0.2 mg/pupa) [Wang et al. (2005) Biotechnol., 69, 385–389] and human granulocyte macrophage colony stimulating factor (0.1mg/pupa) [Chen et al. (2006) J. Biotechnol., 123, 236–247], suggesting that our method can be applicable for isotope-labeling of other proteins in high yields. In replying to the reviewer’s comment, we added sentences with these references (p.7, lines 203-206).

- The spectra obtained with several 15N-13C-double labelled amino-acids and the comparison of what should be expected in the absence of any transamination process or side-chain scrambling. It also would be important to have a measure of the 15N and 13C labelling level for these amino-acids. It would also be important to have a discussion of the labelling process in the context of what is known about the silkworm amino-acid metabolism. 

As noted above, the purpose of this study is not to pursue the details of the 15N labeling and we have already achieved uniform 15N labeling using the silkworm system. Therefore, we will not try selective labeling with the very expensive 15N/13C-double labelled amino acids. However, we suppose that it would be significant to mention the amino acid metabolism of silkworms. Hence, we briefly described it citing relevant papers in the revised manuscript (p.7, lines 203-206).

- A more complete discussion of the advantages/disadvantages and cost of the system with respect with those that have been already published.

We estimate the production costs of isotope-labeled glycoproteins on the basis of our real-life experience of producing 20 mg of rituximab, the mouse-human chimeric IgG1 used as authentic model in this study, One-liter of CHO cell culture yielded 20 mg of IgG. Although the recombinant IgG yield purified from whole-body homogenates of pupae was 0.35 mg/pupa, we have previously demonstrated that antibody production can be improved by a factor of 5 upon co-expression with molecular chaperones (ref). Therefore, 20 mg of antibody could be produced with 13 pupae when co-expressed with the chaperones. Under this circumstance, the mammalian cells and silkworm pupae are almost equal in terms of the costs for producing the isotope-labeled IgG glycoproteins, although depending on labeling modes. However, it should be noted that the silkworm expression systems do not require any expensive equipment such as CO2 incubator. Therefore, the total cost will be less expensive in the silkworm-based method, which also does not take up much space, enabling production of different proteins with different labeling modes in parallel.

In terms of time period for obtaining the products, both systems require approximately 3 to 4 weeks from the start of culture with isotope-labelled metabolic precursors to harvest. However, it generally takes a couple of months to establish a mammalian cell line that can producing recombinant antibody at high expression level. In total, the pupa expression system is excellent in terms of flexibility and efficiency, in contrast to large-scale culture of mammalian cells.

In replying to the reviewer’s comment, we added sentences in the revised manuscript (p.10, lines 251-262).

Reviewer 2 Report

In this manuscript, the authors introduced a method that allows the metabolic labeling of proteins expressed in silkworm pupae system. The conditions of labeling method were investigated and the labeling efficiency was determined to be 80%. The outcome of this study provides an alternative approach of protein labeling in silkworm pupae system. The innovation and experimental design of this study are promising. However, my major concern is the low labeling efficiency. The SILAC method in cell culturing has been well established with a very high labeling efficiency, which can be used for multiplexing. Therefore, please address the significance of this study. There are additional comments as follows:

  1. In Table 1, please specify what are “A40M10”, “A30M10”, and “A20M20”.
  2. The authors mentioned “N-glycans of the pupa-derived IgG glycoprotein were released by hydorazynolysis”. Could it be hydrazinolysis which is used to release glycans?
  3. In Tabel 3, please specify what does “G.U” stand for.
  4. In the whole manuscript, the authors did not show any data of IgG separation and purification. Please add some figures to show the results of each purification step, so that people can repeat this experiment easily.
  5. Have the author compared this method to cell-based expression system? In cell cultures, recombinant proteins can be easily labeled using SILAC method with a very high labeling efficiency. Please explain more about the advantages of choosing such silkworm expression system. A comparison between these two methods is highly recommended, since the authors have already performed CHO cell expression in this study.
  6. Please provide a table listing MRM parameters for all analytes.
  7. Please provide the full MS and MS2 for glycan structural assignment.
  8. The labeling efficiency is only 80%, which is not acceptable. Usually, the metabolic labeling efficiency should be reach 95%. What could be the reason? Are there any ways to improve the labeling efficiency? Please add a paragraph to discuss this issue.

Author Response

Response to reviewer 2

In this manuscript, the authors introduced a method that allows the metabolic labeling of proteins expressed in silkworm pupae system. The conditions of labeling method were investigated and the labeling efficiency was determined to be 80%. The outcome of this study provides an alternative approach of protein labeling in silkworm pupae system. The innovation and experimental design of this study are promising. However, my major concern is the low labeling efficiency. The SILAC method in cell culturing has been well established with a very high labeling efficiency, which can be used for multiplexing. Therefore, please address the significance of this study. There are additional comments as follows:

We thank the reviewer for the constructive comments. We suppose that

  1. In Table 1, please specify what are “A40M10”, “A30M10”, and “A20M20”.

In replying to the reviewer’s comment, the notations of artificial diets were defined in the main text (p.2, lines 80-82).

  1. The authors mentioned “N-glycans of the pupa-derived IgG glycoprotein were released by hydorazynolysis”. Could it be hydrazinolysis which is used to release glycans?

Yes, hydrazinolysis is widely used for releasing N-glycans form glycoproteins. The relevant references were cited in the revised manuscript (p.6, line 158).

  1. In Tabel 3, please specify what does “G.U” stand for.

As per the reviewer’s comments, the definition of the glucose unit (G.U.) was provided in the main text (p.6, lines 161-162).

  1. In the whole manuscript, the authors did not show any data of IgG separation and purification. Please add some figures to show the results of each purification step, so that people can repeat this experiment easily.

We thank the reviewer for the constructive comments. Accordingly, we added the scheme for purification of IgG and its proteolytic fragments along with the SDS-PAGE profiles of the Fab and Fc fragments under reducing and non-reducing conditions (Figure 3).

  1. Have the author compared this method to cell-based expression system? In cell cultures, recombinant proteins can be easily labeled using SILAC method with a very high labeling efficiency. Please explain more about the advantages of choosing such silkworm expression system. A comparison between these two methods is highly recommended, since the authors have already performed CHO cell expression in this study.

We envision the application of this method to NMR analysis, which requires isotope-labelled recombinant proteins in mg quantities. Although the mammalian cells and silkworm pupae are almost equal in terms of the costs for producing the isotope-labeled IgG glycoproteins, the silkworm expression systems do not require any expensive equipment such as CO2 incubator. Therefore, the total cost will be less expensive in the silkworm-based method, which also does not take up much space, enabling production of different proteins with different labeling modes in parallel. Both systems require approximately 3 to 4 weeks from the start of culture with isotope-labelled metabolic precursors to harvest. However, it generally takes a couple of months to establish a mammalian cell line that can producing recombinant antibody at high expression level. Thus, the pupa expression system is excellent in terms of flexibility and efficiency, in contrast to large-scale culture of mammalian cells.

In replying to the reviewer’s comment, we added sentences in the revised manuscript (p.10, lines 251-262).

  1. Please provide a table listing MRM parameters for all analytes.

In this study, we did not used any MS/MS analysis. The 13C enrichment degrees of methionine were estimated on the basis of intensities of the isotope peaks in LC-MS analyses.

  1. Please provide the full MS and MS2 for glycan structural assignment.

In this study, we used the HPLC mapping method but not any MS/MS techniques for identifying the N-glycans. In the revised manuscript, we added a new supplemental figure showing the HPLC profile (Supplemental Figure 2).

  1. The labeling efficiency is only 80%, which is not acceptable. Usually, the metabolic labeling efficiency should be reach 95%. What could be the reason? Are there any ways to improve the labeling efficiency? Please add a paragraph to discuss this issue.

Requirement of labeling efficiency depends on the experiments in which an isotope-labeled recombinant protein is used. We first have in mind its use in NMR studies, in which the 80% 13C enrichment is fine (Of course, the higher, the better). We started rearing silkworm larvae with the isotope-labeled artificial diet at their fifth instar stage. Therefore, the isotope dilution is likely to be ascribed to pre-existing unlabeled amino acids in the silkworm body. An earlier start of rearing with the isotope-labeled diet would improve 13C labeling efficiency but presumably result in a lower production yield. Taking account of such a trade-off, we suppose that an 80% enrichment is a good compromise for measuring heteronuclear NMR spectra. We described these points in the revised manuscript (p.8, line 228-p.9, line 233).

Round 2

Reviewer 1 Report

The authors answered most of the questions addressed at the end of the first reading. They improved their manuscript. I think it is now acceptable for publication with some minor improvements: 1) Several important references concerning uniform and specific labelling methods could be added. 2) The amount of diet necessary to feed a pupae should be given. In the same idea, it would be interesting to know the amount of 13C-Met necessary to produce one typical NMR sample (100 ul at 100 uM, for example).

Author Response

Response to reviewer 1

The authors answered most of the questions addressed at the end of the first reading. They improved their manuscript. I think it is now acceptable for publication with some minor improvements:

1) Several important references concerning uniform and specific labelling methods could be added.

As per reviewer’s comment, two references were cited in the revised manuscript (p.2, line 55).

2) The amount of diet necessary to feed a pupae should be given. In the same idea, it would be interesting to know the amount of 13C-Met necessary to produce one typical NMR sample (100 ul at 100 uM, for example).

We thank the reviewer’s constructive comment. In this study, 2.5 mg of recombinant IgG glycoprotein was subjected to NMR measurements, which was produced with 7 pupae fed with 82 g of artificial diet containing 200 mg of [methyl-13C]methionine. We added this description under Materials and Methods in the revised manuscrupt (p. 5, lines124-126).

Reviewer 2 Report

The authors have addressed most of the comments. The quality of this manuscript is acceptable now. However, please check with the following questions:

  1. Line 155. Please double check the spelling “hydorazynolysis”. I think it should be “hydrazinolysis”.
  2. About the question “Please provide a table listing MRM parameters for all analytes”, the authors responded “In this study, we did not used any MS/MS analysis. The 13C enrichment degrees of methionine were estimated on the basis of intensities of the isotope peaks in LC-MS analyses.” However, in the manuscript, line 146-147, they mentioned “The MRM parameters for each of 35 hydrophilic metabolites were optimized by flow injection analysis.” Please explain.

Author Response

Response to reviewer 2

The authors have addressed most of the comments. The quality of this manuscript is acceptable now. However, please check with the following questions:

Line 155. Please double check the spelling “hydorazynolysis”. I think it should be “hydrazinolysis”.

We apologize for our typographical error, which was corrected in the revised manuscript.

About the question “Please provide a table listing MRM parameters for all analytes”, the authors responded “In this study, we did not used any MS/MS analysis. The 13C enrichment degrees of methionine were estimated on the basis of intensities of the isotope peaks in LC-MS analyses.” However, in the manuscript, line 146-147, they mentioned “The MRM parameters for each of 35 hydrophilic metabolites were optimized by flow injection analysis.” Please explain.

We apologize again for our careless mistake. In fact, we did not used any MS/MS analysis in this study. Therefore, the sentence “The MRM parameters for each of 35 hydrophilic metabolites were optimized by flow injection analysis.” was deleted in the revised manuscript.